# Spark Plasma Sintering of Variable SiC α/β Ratio with Boron and Carbon Additions—Microstructure Transformation

Marek Kostecki *, Mateusz Petrus , Tomasz Płociński and Andrzej Roman Olszyna

Faculty of Materials Science and Engineering, Warsaw University of Technology, 141 Wołoska,
02-507 Warsaw, Poland
* Correspondence: marek.kostecki@pw.edu.pl

**Abstract:** This study investigated the possibility of obtaining dense silicon carbide sinters with the use of a spark plasma sintering (SPS) process by changing the composition of SiC α/β polymorphs in a starting powder mixture. Amorphous boron was used as the basic additive to activate the sintering processes. Some of the compositions were prepared with additional carbon in two different forms: multilayer graphene flakes and carbon black. The well-described effect of the β−α transition in the form of elongated lamellar grains in the sintered structure was confirmed. The obtained sinters were analyzed qualitatively and quantitatively in terms of the microstructure and density. The hardness and the participation of the polytypes in the sinter structures were examined. During the study, SPS sintering allowed us to obtain a material with a density close to the theoretical (relative density of 99.5% and hardness of 27 MPa) without the addition of carbon. It was found that the role of carbon was not limited to the activation of the sintering process. Additional effects accompanying its presence, depending on the initial α/β composition, included grain size reduction and an influence on the transformation kinetics.

**Keywords:** silicon carbide; SPS sintering; SiC β-α transition





## 1. Introduction

Silicon carbide is a remarkable ceramic. It is characterized by high thermal stability and resistance to thermal shock. As a result, it can be part of refractory elements or devices operating at very high temperatures. Among ceramics, it is one of the best heat conductors. The conductivity coefficient for polycrystalline single-phase SiC is 60–100 W/m·K, while for pure single crystals, it reaches 500 W/m·K. It is also an extremely hard (~30 GPa) and chemically resistant material [1]. Thanks to this combination of properties, it has a very wide range of applications. It is suitable for cutting tools, provides excellent ballistic protection, and is used in the metallurgy of non-ferrous materials and heating devices. Currently, the largest amount of research is focused on the semiconducting properties of SiC in electronics in the form of a single crystal or layers that are obtained during gas phase synthesis [2].

For the majority of construction applications, the most important are polycrystalline sinters. The production of these has always presented many difficulties. Due to the covalent nature of the chemical bonds in silicon carbide, high pressure and high temperature are used to bond the powder particles from the very beginning [3]. Therefore, in the early 1970s, additives were used to intensify the sintering process. Acquiring a dense sinter is essential for obtaining the highest mechanical properties, which include hardness, strength and fracture toughness.

There are two main groups of sintering activators: oxide and non-oxide additives [4–6]. The use of oxides, such as $Al_2O_3$, $ZrO_2$ or $Y_2O_3$, as sintering aids results in the formation of a liquid phase during sintering [7,8]. The advantage of this solution is the reduction in sintering temperature and the high fracture toughness of such sinters. Unfortunately,

this is accompanied by a significant decrease in hardness and thermal resistance [9,10]. In turn, the use of non-oxide additives, such as boron and carbon, allow one to obtain sinters with a high relative density and high mechanical and thermal properties, close to the level achieved by the single crystal [11,12]. Research on this group of sintering additives has been ongoing for several decades. On this basis, it can be concluded that boron activates volume mass transport processes by creating a thin glassy phase at the grain boundaries. Moreover, the diffusion of boron into the SiC grains increases the vacancy concentration and diffusivity.

The first hypothesis regarding the role of carbon in the silicon carbide sintering process suggested that it is responsible for the reduction in silicon oxide present on the SiC grain surface, which was done to accelerate the diffusion processes during sintering. However, subsequent studies showed that the oxide layer automatically reduces above 1400 °C, which is a temperature well below the minimum temperature required to start the sintering processes. Subsequent studies on the role of carbon in the sintering process showed that carbon reduces inefficient mass transport through the gaseous phase, binding volatile compounds formed during the decomposition of sintered SiC [5,13]. According to the proposed model, in samples without the addition of carbon, the reactions during sintering can be summarized as follows:

$$SiO_2 + nSiC \rightarrow Si_g + SiO + CO_g + (n-1)SiC$$

whereas in samples with the carbon addition, the reaction can be written as follows:

$$SiO_2 + nSiC + 3C \rightarrow CO_g + (n+1)SiC$$

Mass transfer during the sintering process through the gas phase leads to spheroidization of the pores, and thus, to a reduction in the radius of curvature at the contact perimeter of the two grains. This is a disadvantageous phenomenon because the smaller the radius of curvature, the lower the stresses within the two grains, which is a force that inhibits effective (due to the elimination of pores) diffusion processes. By binding volatile compounds, carbon limits mass transfer through the gas phase, thus promoting efficient mass transport processes.

The role of carbon in the process of sintering a dense SiC is not limited to activating the processes of sintering. It was experimentally confirmed that by increasing the proportion of carbon in the form of graphite to 6%, a reduction in the grain size was observed. Refs. [11,14] Other studies carried out later by the authors indicated that the excess amount of carbon may effectively influence the stabilization of the cubic phase [15]. At the same time, not only its quantity but also its morphological form can strongly affect the manner of densification [16]. The use of carbon in the form of well-dispersed sub-micron equiaxial particles enables one to obtain sinters with a relative density similar to the theoretical density in the broadest scope of weight content (from 0.5 to 1.5 wt%). The use of larger particles and flake morphology made it impossible to achieve high densities.

The additives used have, of course, a very significant influence on the obtained properties. However, without an appropriate sintering technique, achieving a high degree of densification is still difficult. One of the most modern techniques that allows one to significantly improve the properties of SiC sinters is SPS [17]. Thanks to the effective use of the phenomenon of a microscopic electric discharge between the powder particles, as well as the use of external pressure, similar to hot working, the process can be carried out very efficiently by activating diffusion processes and reducing the total time needed for consolidation. In addition, this sintering technique allows the entire sample to be evenly heated, even in the case of materials with a thermal conductivity lower than metals. This was confirmed by one of the first attempts to obtain a SiC sinter using SPS carried out by Tamari and his team [18]. Subsequent work confirmed that the use of the SPS technique can effectively eliminate the need for additives [19,20]. Finally, fully dense silicon carbide

specimens with fine equiaxed grains (average grain size is 1.28 μm) were fabricated using SPS technology at 2050 °C in the absence of sintering additives [21].

An often-overlooked issue in the sintering process is the crystallography of silicon carbide. However, it turns out that the initial composition of silicon carbide in terms of the polymorphic form, as well as the morphological form of SiC, may also be of key importance for achieving the desired microstructure.

More than 250 polytypes of silicon carbide have been discovered [22]. The polytypes of silicon carbide can be classified into two phases: α and β. The β phase is metastable at any temperature and undergoes a transformation into an α phase under the influence of various factors (temperature, protective atmosphere, dopants) [23,24].

The phase transitions that occur in the process of sintering under the influence of temperature are not desirable phenomena since they preclude control of the forming microstructure. Years of research enabled Ramsdel and Kohn to create a theory in which phase crystallization is the result of joining a set of atoms from the gas phase to a crystal surface. The arrangement of atoms is dependent on temperature [25–27]. Several factors that stabilize certain phases were specified, e.g., boron stabilizes the 6H type, whereas alumina is conducive to the creation of the 4H type [28,29]. Some results suggested that the precise control of phase transformation in SiC ceramics and their mechanical properties could be achieved through annealing with or without pressure [30].

In practice, the microstructure of SiC obtained from α-SiC powder consists of fine equiaxed grains. Moreover, it is usually more brittle than SiC obtained from β-SiC powder, which, in turn, consists of coarse, elongated grains as the result of the growth accompanying the β–α phase transition. Larger elongated grains increase the fracture toughness of SiC via crack bridging or crack deflection. Therefore, the most common method of adjusting the microstructures and mechanical properties of liquid-phase-sintered SiC ceramics is to vary the α-SiC/β-SiC ratio of the starting powders [31].

As shown in the above considerations, polymorphic transformations are an indispensable element of the silicon carbide sintering process when it comes to sintering powder with a cubic structure. The microstructure achieved by them may be important for both the sintering process itself and for the properties of the sinter. As the transformation phenomenon is well-researched and documented, the main goal of this experiment was to investigate the influence of the share of α-SiC and β-SiC powders subjected to the SPS sintering process and to determine the optimal composition of the mixture. The additives were used to indicate whether they would be necessary to obtain a sinter with optimal properties. The basis for the evaluation of the sintering process in the presented study was the relative density and the quantitative description of the microstructure, taking into account the presence of different polytypes.

## 2. Materials and Methods

For the preparation of the mixtures, silicon carbide powders of hexagonal α-SiC and regular β-SiC purchased from Alfa Aesar (Alfa Aesar, Ward Hill, MA, USA) were used. Both were characterized by a chemical purity of 99.8%. The purity and phase composition were confirmed using XRD tests. Both the α-SiC and β-SiC powders were also characterized by a fairly large particle size distribution and the average particle diameters were 1.3 and 0.8 μm, respectively. The morphologies of the powders used are shown in Figure 1.

Additives of 0.3 wt.% boron and 1 wt.% carbon were used. Since the authors' previous research clearly suggested that the morphological form of carbon itself may be important for the sintering process [16], two types were used: (1) carbon black (CB) with a purity above 99% and an average particle size of 80 nm and (2) multi-layered graphene (MLG) with 99.9% purity (Graphene Laboratories Inc., Ronkonkoma, NY, USA), a flake thickness of 8 nm and an average diameter of 5 μm. Amorphous boron of 96% purity (International Enzymes Limited, Hampshire, UK) was also added to each sample in the form of a dark brown powder with an average particle size of 350 nm.

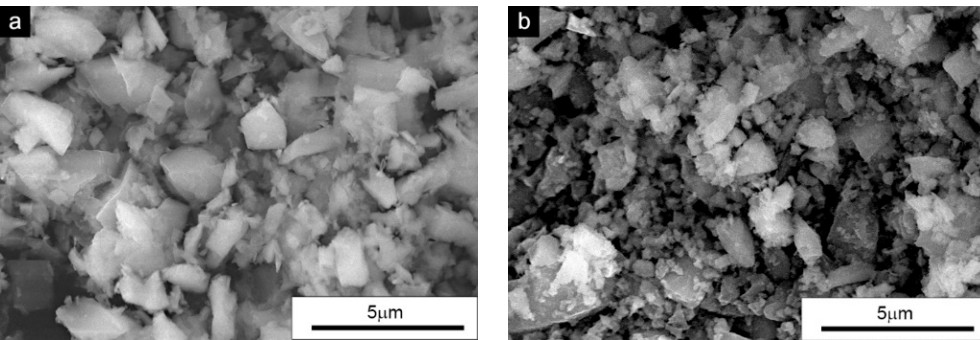

**Figure 1.** The morphologies of the starting powders: (**a**) α-SiC and (**b**) β-SiC.

Sintering was performed by establishing the optimal parameters based on the literature [20] and the authors' previously researched optimization process. The powders were consolidated using the spark plasma sintering (SPS) method. A sintering temperature of 1900 °C and a sintering time of 30 min were used. The heating and cooling rate of the system was 100 °C/min. Sintering was carried out in a die with a diameter of 20 mm and a punch force of 16 kN, which translated into a pressure of 50 MPa, in vacuum conditions. As a result of this process, cylindrical sinters with d = 20 mm and a height of 5 mm were obtained. Before further testing, the graphite layer that remained on the surface of the obtained samples after taking them out of the matrix had to be removed. The share of ingredients in the initial mixture and all variants of the prepared powder mixtures are presented in Table 1.

**Table 1.** Weight fractions of the prepared powder mixtures with additives.

| Sample Name | Composition of Starting Powder α-SiC/β-SiC Ratio wt.% | | | | | Sintering Aids wt.% |
|---|---|---|---|---|---|---|
| **SiC** | 0/100 | 20/80 | 50/50 | 80/20 | 100/0 | 0.3% boron |
| **SiC-G** | 0/100 | 20/80 | 50/50 | 80/20 | 100/0 | 0.3% boron, 1% MLG (graphene) |
| **SiC-CB** | 0/100 | 20/80 | 50/50 | 80/20 | 100/0 | 0.3% boron, 1% carbon black |

Apparent density, defined as the ratio of a material's mass to its total volume, including pores, was measured using the hydrostatic weighing method, which is based on Archimedes' principle. Sample weight measurements were made on a RADWAG WPS 360C analytical balance with a measuring accuracy of 0.001 g, as follows: dry, after saturation with water in the air, and after saturation and immersed in water. In order to saturate the samples, they were boiled in 200 mL of water for 1 h with the addition of 1 mL of surfactant to facilitate the penetration of water into the open pores of the material. Before weighing, the water film was gently removed from the samples.

The hardness of the samples was measured using the Vickers method with a Future-Tech FV-700e hardness tester with a load of 5 kg.

Observations of the microstructure of sinter fractures were made using a HITACHI S-5500 scanning electron microscope (Hitachi S-5500 In-Lens Field Emission Scanning Electron Microscope). The slides were observed using a low-angle technique with a voltage of 2 kV to obtain good indicative contrast. All presented microstructures came from fragments of fractures.

The phase analysis of the samples was performed using a Bruker D8 ADVANCE X-ray diffractometer; Kα Cu radiation (λ = 0.154056 nm) was used in the range of 2θ angles between 10° and 120°.

The stereological analysis carried out to characterize the shape and mean size of SiC grains in the microstructures was performed in the NIS-BR program provided by NIKON.

Two parameters of the microstructure were investigated: equivalent grain diameter $d_2$, which is defined as the diameter of a circle with an area equal to the surface of a given grain, and grain elongation coefficient, which is defined as the ratio of the longest distance between two parallel tangential grains (the so-called maximum Feret diameter) to the shortest distance between two grains with parallel tangents to the grain (the so-called minimum Feret diameter).

The type of polytypes in the studied SiC crystals was determined using the backscattered electron diffraction (EBSD) method. Data acquisition and EBSD analysis were performed by using a scanning electron microscope (Hitachi SU70 Field Emission Scanning Electron Microscope) and Bruker e-flash HD EBSD system. The acquired data were analyzed by using Esprit and Channel 5 software. For the data acquisition, the sample surface was polished mechanically and then additionally cleaned by using a broad ion beam milling system (Hitachi IM4000). The EBSD technique is not the best one for determining phase analysis, especially when we have to distinguish the different polytypes of the same material; therefore, the phase list was determined according to the results obtained from the XRD analysis. This approach gave us much better confidence in the obtained results.

## 3. Results

Since the SiC observed using an electron microscope showed a very clear indicative contrast while using appropriate parameters of the electron beam, it was possible to trace the changes in the structure, including the morphology of the grain and defects in the form of pores and inclusions of other phases. All these elements are discussed on the basis of the photos attached below (Figures 2–4). The photos were taken directly of the fractures because their transcrystalline nature allowed for obtaining extremely smooth surfaces at the observed scale. The described procedure allowed us to avoid tedious preparation, including grinding, polishing and thermal etching in a vacuum while preventing the formation of artifacts.

The selected microstructures obtained in the sinters with different initial powder compositions differed significantly in terms of grain morphology. Figure 2 shows SEM pictures of the sinter microstructure from a powder that contained only the β-SiC phase. According to the literature data, a microstructure consisting of elongated (plate-like) grains is the result of grain growth during the β-SiC → α-SiC phase transformation. This hypothesis was confirmed by the fact that the initial grain shape of the β-SiC powder before the process was close to equiaxial (Figure 1). Such a structure was also characterized by high porosity, which was the reason for the low density of this sample. The privileged direction of crystal growth during sintering was random because the initial grain/particle orientation of the powder was random. New grains growing in a privileged direction formed a "grid" with numerous voids. The darkest areas between the grains were the pores and, in the case of carbon addition (Figure 2b,c), residual carbon particles.

It can be observed that the carbon particle agglomerates formed structures with sizes up to several micrometers. Graphene flakes with agglomerates reaching 6 μm are particularly unfavorable. While in the case of carbon black, obtaining a good dispersion is easier due to its morphology, graphene in the form of flakes is an additive that in such amounts usually leads to a deterioration of the structure [32], and its optimal amounts added in composite materials to improve mechanical properties have a small range of fractions of a percent. Although desirable in the sintering of SiC, the presence of carbon in the analyzed case led to a reduction in density, microstructural inhomogeneities and possibly a deterioration of mechanical properties.

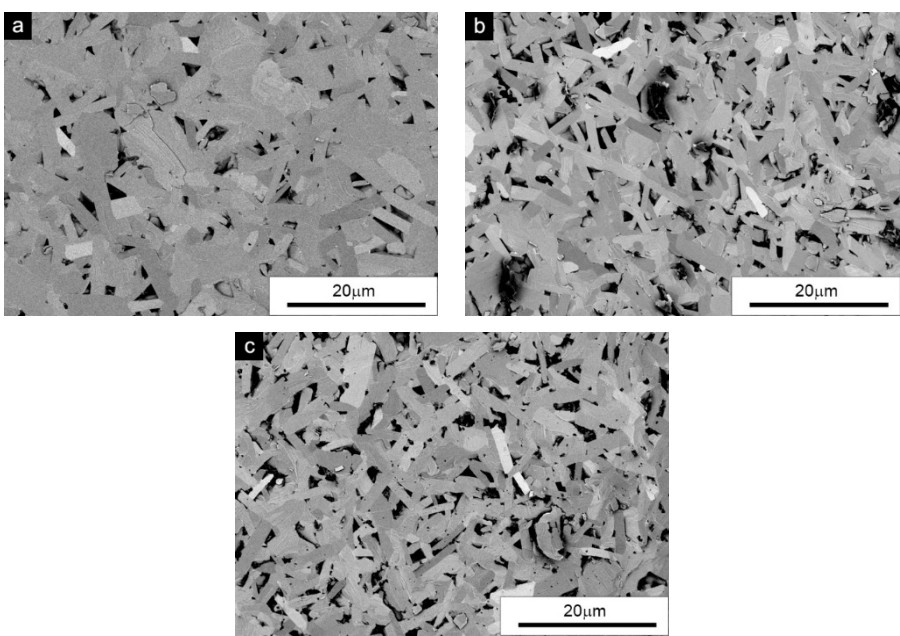

**Figure 2.** Microstructure of SiC sinters' fractures for the starting powders with an $\alpha/\beta$ ratio of 0/100: (**a**) carbon-free SiC, (**b**) SiC-G graphene addition and (**c**) SiC-CB carbon black addition.

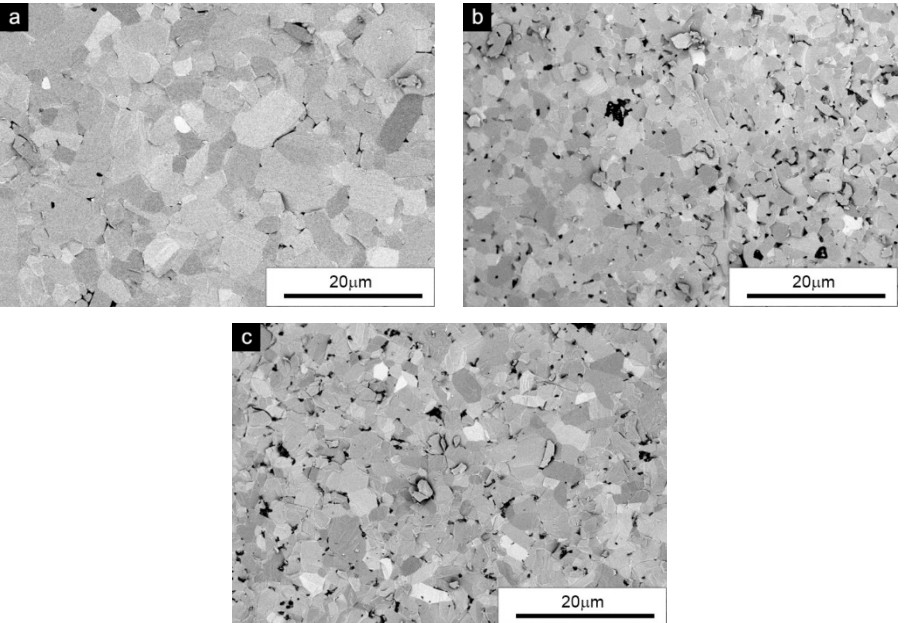

**Figure 3.** Microstructure of SiC sinters' fractures for the starting powders with an $\alpha/\beta$ ratio of 50/50: (**a**) carbon-free SiC, (**b**) SiC-G graphene addition and (**c**) SiC-CB carbon black addition.

Qualitative observations of the microstructure and subsequent analyses of the grain shape clearly showed that even a 20% share (in the examined case) in the initial mixture of hexagonal particles completely modified the obtained sintered microstructures. Along with increasing the proportion of the $\alpha$-SiC phase in the powder mixture, the morphology of the sinter grain changed in the equiaxial direction. In Figure 3a, the microstructure of the sinter with a phase ratio of 50/50 (without the addition of carbon) was characterized by a barely perceptible porosity with nano-sized pores and mainly equiaxed grain morphology. Preparations with the addition of carbon are characterized by a clearly visible porosity and carbon residues in the pores. However, no agglomerates of graphene flakes were

observed as in the case of 0/100 samples. Small chipping was observed on the fracture surface. The fracture surfaces were transcrystalline, just like the previously described case. The qualitative evaluation of the sinters showed a slightly smaller and more homogeneous grain when carbon was added (Figure 3b,c).

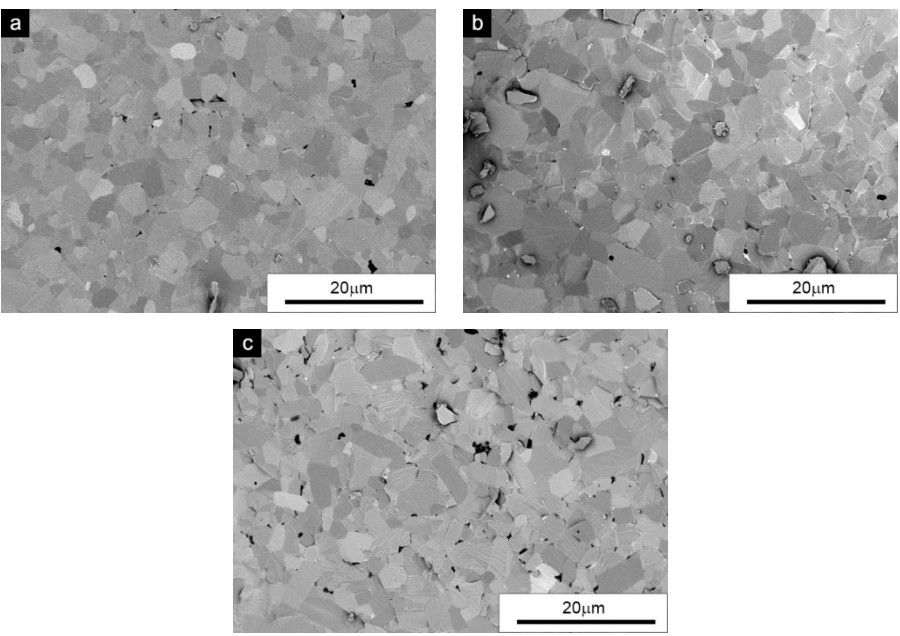

**Figure 4.** Microstructure of SiC sinters' fractures for the starting powders with an α/β ratio of 80/20: (**a**) carbon-free SiC, (**b**) SiC-G graphene addition and (**c**) SiC-CB carbon black addition.

The situation slightly changed with the dominant share of α-SiC phase particles in the powders mixture (α/β ratio of 80/20—Figure 4). For these sinters, very similar microstructures were observed and were characterized by a small number of pores and a fairly homogeneous fine grain, regardless of the type of carbon additives used. The results of the density test (Figure 5) presented below confirmed that for the sinters with the addition of carbon, such a phase ratio gave optimal compaction and may translate into an improvement in mechanical properties. The sinters without the addition of the β-SiC phase in the mixture of starting powders did not differ in terms of the microstructure, obtaining very satisfactory levels of relative density in the range of 98–99%.

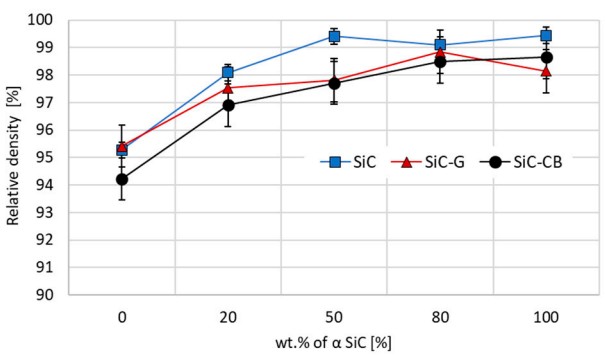

**Figure 5.** Relative densities of the SiC sinters.

Based on the density results (Figure 5), it can be said that most of the obtained sinters, except for those that were not part of the initial mixture of α-phase powders (α-SiC/β-SiC ratio 0/100), were characterized by a high relative density above 97%, which demonstrated the proper selection of sintering parameters. Quite a surprising observation concerned sinters with no carbon addition. These showed the highest density in general,

while the initial composition α/β ratio of 50/50 reached a density that was 99.5% of the theoretical density.

Among sinters with the carbon addition, both graphene and carbon black generated the highest density with a small proportion or none of the β-SiC phase. Carbon black had the greatest impact on the density of the sample with an α/β ratio equal to 100/0; however, as was the case with other sinters with this addition, the density values were lower compared with the samples without CB or MLG.

Although the presence of carbon in the proposed amount did not improve the density, its influence on the grain size was observable. For all compositions containing a high proportion of β-SiC, the grain size of the carbon black sinters was lower, ranging from 23% to almost 50% (Figure 6). Since the transformation from β−α was associated with grain growth, the effect of fine-dispersive carbon with its significant share may be associated with nucleation of the new phase grains and/or limitation of the growth of newly formed alpha grains.

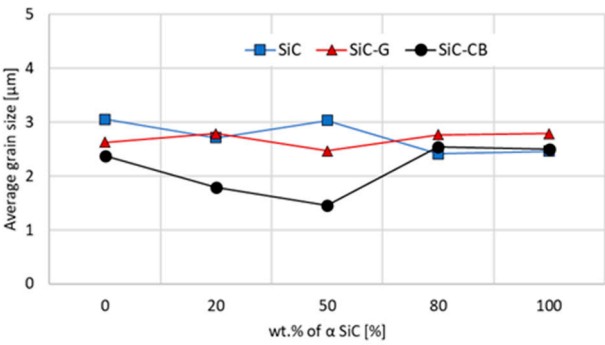

**Figure 6.** Average grain size of SiC sinters.

Regardless of the carbon addition, the SiC sinters produced from the α-SiC starting powder were characterized by grains with a low elongation factor (Figure 7). Having the β phase present in the starting powders resulted in the appearance of elongated grains in the microstructure, as previously described in the analysis of the SEM images. The elongation coefficient was basically the same and reached a value close to 3 for all sinters. The addition of the α phase in the amount of 20% by weight caused a significant decrease in the mean elongation of the grains (Esf ≤ 2) and it seemed that carbon did not have a significant influence on this process. On the other hand, a further increase in the proportion of α in the sample resulted in a very slight decrease in the elongation coefficient. In the case of the carbon black sinters, it remained relatively equal for all formulations containing α-phase powders in the starting composition.

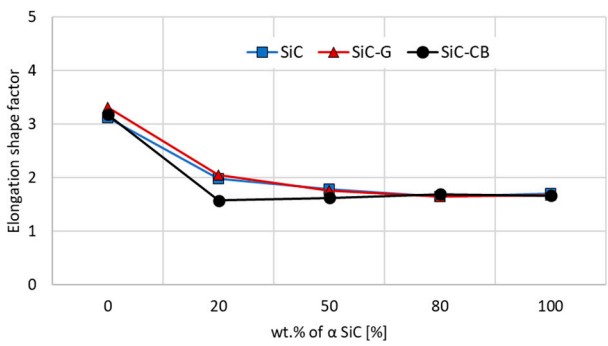

**Figure 7.** Grain elongation shape factor of SiC sinters.

The hardness results (Figure 8) correlated very well with the sintered density, and in classical terms, they confirmed the theory that the strongest factor influencing hardness

was a high density. Sinters made exclusively of the β-SiC powder, regardless of whether carbon was added to them, were characterized by a very low hardness. Even a drastic reduction in the grain size (about 50%) did not improve the situation of the 50/50 sinters characterized by a 1.5% lower relative density than the samples without the addition of carbon. The highest hardness values were obtained for the 50/50 sinters without carbon (27 GPa) and 80/20 with graphene (26 GPa), which were very good in comparison with the literature data [1]. On this basis, it can be concluded that the addition of graphene or carbon black was not the key factor that ensured the improvement of the final hardness of the sintered silicon carbide when the powders were sintered with the SPS technique. In the analyzed variant, due to the generated porosity and the observed lack of homogeneity of compaction, the presence of carbon could be assessed as undesirable.

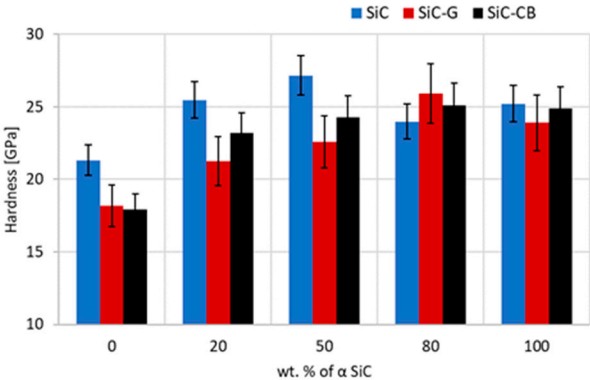

**Figure 8.** Hardness of the SiC sinters.

From the XRD tests that were carried out (Figures 9 and 10), it was not possible to clearly indicate whether there were cubic phase grains/regions left after the sintering process. The XRD examination carried out on the sinter in which the starting material was 100% β-SiC powder with 3C polytype allowed us to identify some hexagonal polytypes, i.e., 6H and 4H, and these were the dominant phases (Figure 9). In addition, 15R and free carbon were also identified in the sinters. The great similarity in the symmetry of crystal structures caused the peaks of the characteristic hexagonal (6H, 4H) and cubic phases (3C) to coincide and it was not possible to uniquely identify polytypes with this method.

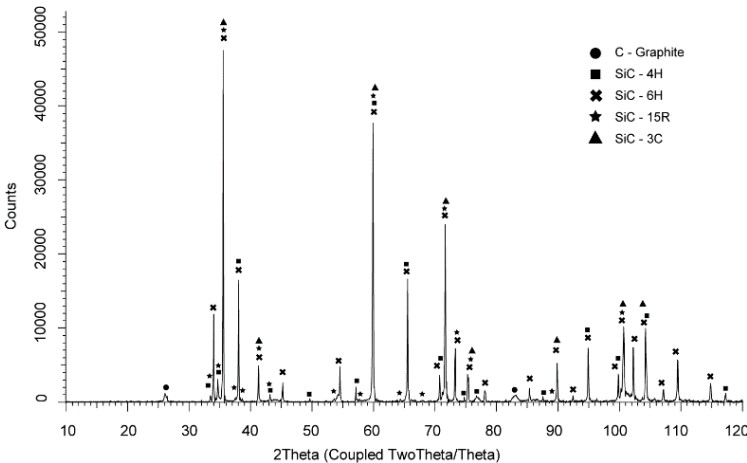

**Figure 9.** XRD pattern of the 0/100 SiC sinter with graphene addition.

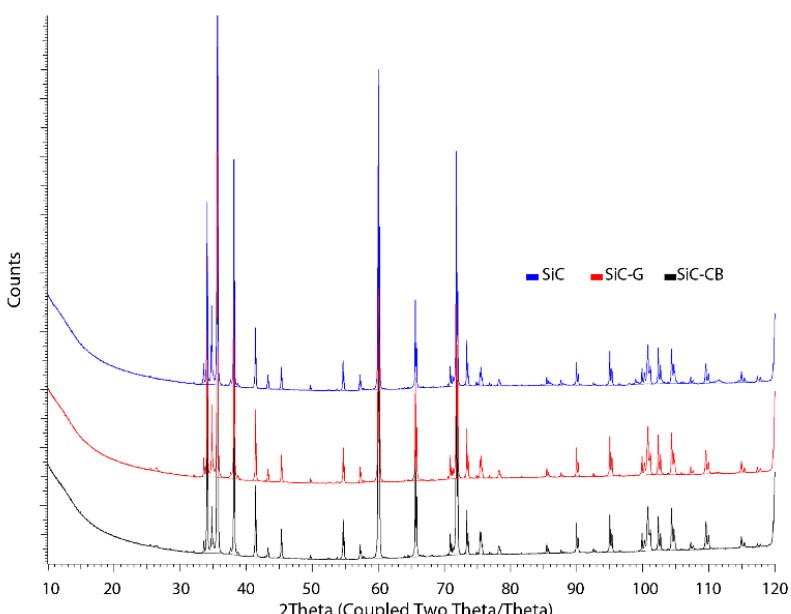

**Figure 10.** XRD patterns comparison of the 3 different 50/50 sinter types.

The features of the diffraction patterns for the sinters with the same starting composition (Figure 9) of 50/50 SiC without and with the addition of carbon suggested that the carbon did not have a significant effect on the final phase composition. The diffraction patterns basically matched and the intensities of the peaks were similar. It cannot be ruled out that carbon may have affected the nucleation rate of new grains, which would explain the significant grain refinement, especially when its source was made of very fine particles. In order to confirm the phase change and link the observed microstructure with specific varieties of polytypes, EBSD tests were performed, which allowed for assigning individual grains to a specific spatial symmetry or orientation.

The EBSD tests were performed for samples of the 20/80 series in which a high share of the β-SiC phase, namely, 80%, may indicate differences in the level of 3C polytype transformation and the shaping of the microstructure resulting from the carbon content. An exemplary image of the structure with an indication of the affiliation of individual grains or regions to a specific crystallographic system is shown in Figure 11.

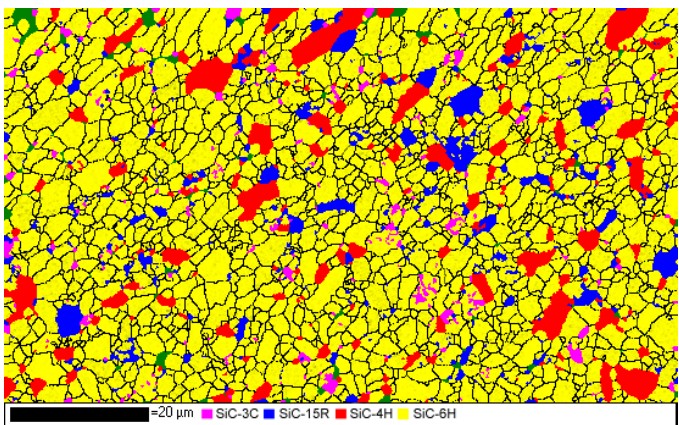

**Figure 11.** Example of the EBSD pattern of the 20/80 SiC sinter SiC-G.

Table 2 presents the quantitative results of the study taking into account the percentage share of the polytype in the structure. In the case of samples without carbon addition in the starting mixture, the 3C SiC polytype was not observed. This showed that a complete

beta–alpha transition took place. The samples containing carbon in their initial composition were characterized by the presence of about 6% and 9% of unchanged polytype 3C areas in the structure. Statistically, more areas were found in the preparations in which finer and better-dispersed carbon in the form of carbon black was used. Since the presence of carbon favors the transformation of polypropylene 6H, its share in all tested samples was much greater (~60%) than the 4H polytype, while the 15R polytype was observed at levels ranging from 15 to 24%.

**Table 2.** Participation of individual polytypes in the structure of SiC 20/80-type sinters.

|       | 3C    | 15R    | 6H     | 4H     |
|-------|-------|--------|--------|--------|
| SIC   | 0.00% | 24.03% | 59.68% | 16.29% |
| SIC-G | 6.17% | 22.48% | 59.57% | 11.78% |
| SIC-CB| 8.82% | 15.27% | 65.94% | 9.98%  |

## 4. Discussion

The thermodynamic conditions of the sintering process, namely, the pressure and temperature, at which the synthesis was carried out favored the β–α transition, in accordance with the studies cited [33,34]. Therefore, in the case of samples consisting exclusively of the beta phase, we observed a structure with elongated grains that confirmed the transformation. It seemed that in this case, carbon was not of great importance for the ongoing transformation. The observed particle size was slightly smaller in the samples that contained carbon, but their elongation remained the same. The transformation and growth of the grain must have occurred fairly quickly because it did not fully densify and there was a large number of voids between the elongated grains. On the other hand, carbon accumulated in the empty spaces, which was clearly visible in the samples' microstructures.

The first symptoms of microstructural changes resulting from the addition of carbon were noticed in samples with the addition of the 20% alpha phase. The obtained sinters were characterized by a significant reduction in grain elongation, as well as the addition of carbon, especially in the form of highly dispersed fine CB particles, which led to a grain size reduction of more than 20%. This effect was even more noticeable for samples with an equal share (50/50) of β and α phases in the initial mixture, where the grain size decrease reached 50%. In principle, a further increase in the alpha phase content equalized all microstructure parameters, and sinters, with and without carbon, only differed in porosity. Surprisingly, compositions without carbon addition were characterized by the lowest porosity.

The literature indicates three possible mechanisms for the β–α transformation. The first concerns the movement of partial dislocations, creating alignment errors that migrate and change the polytype [35]. The second mechanism assumes that the new phase seeds are formed in the process of evaporation and condensation [36], while the third suggests nucleation inside the grain volume with a favorable crystal structure [37] (seed grains of a favorable crystal structure grow to consume the surrounding matrix). It is difficult to clearly indicate which of them will play a dominant role in the formation of the microstructure under the proposed conditions. However, it can be assumed that the previously presented carbon effect will take place simultaneously with the transformation, that is, carbothermal reduction from the surface of the native $SiO_2$ oxide layer. Carbon binds the volatile compounds formed as a result of the evaporation of the surface layers of silicon carbide grains, and thus, intensifies the self-diffusion [10,12]. Since during the SPS process, spark discharges can lead to local evaporation of the material (much easier than in pressure sintering where an appropriate temperature of the sintering process is required), the suggested grain size reduction mechanism may take into account the formation of new crystallization seeds in the presence of carbon, Si and $SiO_2$ vapors. At the same time, the presence of the alpha phase, which is thermodynamically stable under sintering conditions, causes the beta phase particles after transformation to have limited space for free growth. The increasing amount of the alpha phase in the sinter only enhances this effect. This suggests that the decrease in the average particle size of the obtained composites may

have been the result of both growth limitation and the formation of new nuclei due to the presence of well-dispersed carbon.

It cannot be clearly indicated on the basis of the performed studies whether the carbon slowed down the β–α conversion in some way, although the EBSD studies carried out for the 20/80 samples indicated a greater amount of beta phase remaining in the sinters when additional carbon was present. As shown by the EBSD studies, the structure of the sinters containing carbon remained in the range of 6% to 8% of the 3C phase. By changing the chemical composition of the vapors of atoms formed during sintering in the spaces between the grains, carbon also changes its spatial arrangement, which perhaps promotes the formation of the beta phase. It cannot be ruled out that the crystallization of the volatile $SiO_2$ decomposition products in the presence of excess carbon took place in the form of the 3C polytype [38], extending the time of complete conversion of all grains beyond the scheduled sintering time.

There is no doubt that the kinetics of transformation and even the grain growth mechanism during sintering will differ depending on the content of individual phases (α/β ratio). To fully understand the mechanisms, extended research is required that takes into account variable thermodynamic conditions and sintering time, but above all, high-resolution TEM research is required that shows differences in the structure during the transformation.

A very important aspect not discussed at all in most of the works where the sinters were produced using the SPS method is the diffusion of carbon from the matrix in which the synthesis process was carried out. Several effects of a spark discharge favor its occurrence in the surface layers or even in the entire sinter volume (depending on the process conditions and the type of material to be sintered) and have been observed for both metallic and ceramic materials [39]. It should be assumed that the amount of carbon in the research can significantly increase, at least in the near-surface areas, which explains its presence in the pores of the material after the sintering process.

## 5. Conclusions

The research confirmed the possibility of producing dense SiC sinters by using an SPS sintering process. The best results were achieved for samples containing only boron as the sintering aid, although the participation of carbon in the sintering process cannot be ruled out because of diffusion from the carbon die.

The best density and hardness were obtained for the sinters with a proportional amount of the α and β phase in the starting powder. The material obtained had a relative density of 99.5% and a high hardness of 27 MPa. Thus, there was an optimal proportion of α/β phases to obtain a homogeneous microstructure that was free of pores.

The use of only β-SiC particles as the starting material in the experiment led to a microstructure with highly elongated grains. The phase transformation is well described in the literature and it was shown that the type and amount of carbon used are irrelevant to its occurrence. The presence of 20% of the α-SiC phase in the initial composition led to a significant elimination of the grain elongation effect. Although the elongation factor decreased, the grains were still slightly elongated.

The grain size for compositions up to 50/50 was strongly dependent on the carbon morphology. The fine dispersion of carbon contributed much more to the reduction of the sinter grain size than MLG flakes.

The research showed that the role of carbon was not limited only to an additive that influences the compaction process and the final size of the sinter grains. If we take into account the initial crystallographic structure of the SiC powder and its changes during the sintering process, carbon was shown to actively participate in this transformation and could stabilize the beta phase. On the basis of the presented results, it was not possible to clearly indicate the mechanism of this stabilization.

**Author Contributions:** Conceptualization—M.P. and M.K.; original draft preparation, review and editing—M.K. and M.P.; methodology and investigations—M.K., M.P. and T.P.; project administration—A.R.O. All authors read and agreed to the published version of the manuscript.

**Funding:** This research received no external funding.

**Institutional Review Board Statement:** Not applicable.

**Informed Consent Statement:** Not applicable.

**Data Availability Statement:** No publicly accessible repository for this research. The data presented in this study are available on request from the corresponding author.

**Conflicts of Interest:** The authors declare no conflict of interest.

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
