# Peer review of "Spark Plasma Sintering of Variable SiC α/β Ratio with Boron and Carbon Additions—Microstructure Transformation"

_ceramics, doi:10.3390/ceramics5040089_

Round 1

Reviewer 1 Report

The authors investigated the synthesis of silicon carbide sinters. They studied how the final material will depend on the typo of SiC polymorph used as precursor. This is a very relevant study since the crystalline phase of SiC precursors is usually ignored in sintering processes. The manuscript focus on the characterization of the microstructure morphology of samples prepared for variable relative concentrations of alpha- and beta-SiC polymorphs. The results show that there is an optimal proportion of α/β phases to obtain a homogeneous microstructure free of pores. I recommend publication of this manuscript in ceramics without further revision.

Very minor points:

1.       Page 9 – SPS – please, define acronym in the first appearance. I understand it is obvious from the title, but it is a good practice to define it.

2.       (Introduction) Line 90 – Consider adding some sentences explaining the basic principle of SPS technique.

3.       Line 144 – “The morphology od used powder…” – Please revise sentence.

Author Response

Thank you very much for the positive review and all comments that will improve the quality of the research presented. Below, I would like to refer to the detailed comments contained in the review.

Reviewer 2 Report

The paper reports some experimental results on the effects of the composition of SiC α/β polymorphs in starting powder mixture. The promising density and hardness were obtained for the sinters with a proportional amount of the α/β phase in the starting powder. The high relative dense SiC ceramics with a 99.5% relative density and a high hardness of 27 GPa was obtained. The experimental program is well conducted. While there are sone critical issues that should be addressed before publication in the Journal of Ceramics.

1. The authors are suggested to revise the abstract session. Some importent results should be presented clearly in the absract for easy understanding the main contribution of the work.

2. The grain size distribution should be given for all of the samples. They are important for explaining the properties of the hadness.

3. The authors are suggested measure the thermal conductivity of all samples and discuss it in detail. It should be interesting.

Author Response

(The authors gave the same response as above.)

Round 2

Reviewer 2 Report

The authors responsed the comments. It could be acceppted for publication in the journal of Ceramics.